# Simulation and Experimental Study on Wear of U-Shaped Rings of Power Connection Fittings under Strong Wind Environment

**DOI:** 10.3390/ma14040735

**Published:** 2021-02-04

**Authors:** Songchen Wang, Xianchen Yang, Xinmei Li, Cheng Chai, Gen Wang, Xiaohui Wang

**Affiliations:** College of Mechanical Engineering, Xinjiang University, Urumqi 830000, China; wsc1996@163.com (S.W.); yang92339233@163.com (X.Y.); wang615gen@163.com (C.C.); wgqiuzhi@163.com (G.W.); m17550333191@163.com (X.W.)

**Keywords:** wear, surface morphology, surface analysis, finite element modelling, wear modeling

## Abstract

The objective of this study was to investigate the wear characteristics of the U-shaped rings of power connection fittings, and to construct a wear failure prediction model of U-shaped rings in strong wind environments. First, the wear evolution and failure mechanism of U-shaped rings with different wear loads were studied by using a swinging wear tester. Then, based on the Archard wear model, the U-shaped ring wear was dynamically simulated in ABAQUS, via the Umeshmotion subroutine. The results indicated that the wear load has an important effect on the wear of the U-shaped ring. As the wear load increases, the surface hardness decreases, while plastic deformation layers increase. Furthermore, the wear mechanism transforms from adhesive wear, slight abrasive wear, and slight oxidation wear, to serious adhesive wear, abrasive wear, and oxidation wear with the increase of wear load. As plastic flow progresses, the dislocation density in ferrite increases, leading to dislocation plugs and cementite fractures. The simulation results of wear depth were in good agreement with the test value of, with an error of 1.56%.

## 1. Introduction

Xinjiang plays an important role in the "One Belt And One Road" strategy and the "west–east power transmission" project in China, so the state grid of China has increased the construction of Xinjiang Ultra High Voltage (UHV) transmission lines [1]. However, the climate and environment in the region where Xinjiang UHV transmission lines pass is complicated. Taking the main section of the Xinjiang 750 kV transmission line as an example, it is very prone to the occurrence of gale force 6 or above winds in April–May. Moreover, the annual average number of strong wind days above gale force 8 is more than 100 days, and the maximum wind velocity can reach 44 m/s. In strong wind environments, it is common to find faults such as wear, wind deviation, and pollution flashover of transmission lines. Especially, when the wind force exceeds level 6, the frequency of these faults will be significantly increased, which not only causes tremendous property losses, but also causes huge hidden danger to the safe and stable operation of transmission lines [2,3,4].

In addition to all kinds of static loads, such as the weight of wires, insulators, and other components, the connection fittings are also affected by factors such as breeze vibration, conductor galloping, ice covering, sub-span oscillation, and sleet jump of conductors [5]. According to the data, the 750 kV Tu-ha transmission line had only been in operation for just over a year, and a large number of connection fittings had been worn out and failed, far below normal service life, and which caused direct economic losses of about 10 million yuan. Yang [6] clarified the law of influence of wear times on U-shaped rings. Deng et al. [7] indicated the wear characteristics of U-shaped rings with different concentrations of sand. The above scholars all studied the wear of the connection fittings through experimental methods, with the maximum wear number of up to 250,000 times, and a wear frequency of 1 Hz. This takes a lot of time, manpower, and financial resources. Due to the development of computer technology, the numerical simulation method has become a new means of wear research, which can be used for in-depth study of wear mechanisms, and the prediction of wear development trends [8]. A.L. Mohd Tobi et al. [9] used a cylindrical plane model to simulate fretting wear. Basavaraj Kanavalli [10] utilized the generalized Archard wear model to write the Umeshmotion wear subroutine, to simulate disk rolling/sliding wear depth. Rajesh A M et al. [11] carried out wear testing on a composite material with a pin–disc wear tester, and then simulated the wear depth with finite element software. The test value was in good agreement with the simulated value. P. Arnaud et al. [12] proposed a new finite element energy friction model to simulate the wear surface profile of Ti-6Al-4V. Liu [13] simulated the surface ablation of a carbon–carbon composite based on the adaptive mesh update technology. Based on the above research content, the wear subroutine for connection fittings was written, and the wear simulation was carried out, which could not only greatly shorten the research time, but also create a new method for the study of the wear of connection fittings.

The wear test of the U-shaped rings of power connection fittings was carried out by using the swing wear tester to obtain the wear failure mechanism and damage law. Then, based on the generalized Archard wear model, Fortran language was used to write a wear subroutine for connection fittings, to simulate the wear depth of U-shaped rings dynamically.

## 2. Experimental Details

The wear tests were performed on a swing wear tester, in which the effect of wind was simulated by swinging from side to side [14]. A weight was pulled by a motor to change the wear load. The U-shaped ring swung from left to right by crank-rocker, as shown in Figure 1a. The diameter of the ring was 20 mm, and the radius of the arc at the bottom bend was 25 mm (Figure 1b)). Table 1 lists the chemical composition of the U-shaped rings. As shown in Table 2, according to the designed parameters of the Tu-ha Line II, it can be seen that most of the wire load on the connection fittings is between 4000–8000 N, and the wind deflection angle is about 60°. Therefore, wear loads were selected for 4000 N, 6000 N, and 8000 N. The U-shaped ring’s maximum swing angle from left to right was set as 30°. The upper U-shaped ring was fixed, and end faces were subjected to wear load. The lower U-shaped ring oscillated around its own axis by 30°, and the swing frequency was 1 Hz [15]. Three tests were carried out under the same test conditions, and the wear data were the average of the three test results.

The diameter of U-shaped ring was measured by Vernier caliper (measurement accuracy: 0.02 mm) (Meinaite, Shanghai, China) before and after the test. The wear of the U-shaped ring was determined by the residual diameter. The hardness of the U-shaped ring was measured by a Microvicker’s hardness tester (HXD-1000TB, Shanghai, China), with a load of 200 g-force (HV_0.2_). The hardness was the average value of three measurements. The wear scars of the U-shaped ring were observed and analyzed using a scanning electron microscope (SEM) (LEO-1430VP, Zeiss, Jena, Germany). The chemical composition of the U-shaped rings worn surface was detected via energy dispersive X-ray spectroscopy (EDS, Zeiss, Jena, Germany). Metallographic preparation of the U-shaped ring was performed by using abrasive SiC papers (P120, P400, P600, P800, P1200, and P2000) (Suzhou, China) to remove the contaminated layers. Then, the samples were polished with diamond paste, and etched with nital (4% HNO_3_ in ethanol) to observe the metallographic structure through optical microscopy (OM) (STM7-BSW, Olympus, Tokyo, Japan). The U-shaped ring was thinned by electrolytic double-injection thinner (TenuPol-5, Struers, Ballerup, Danmark). Then, the wear edge microstructure of the U-shaped ring was observed by transmission electron microscopy (TEM) (JEM-200CX, NEC, Tokyo, Japan).

## 3. Results

### 3.1. Wear Behavior

The diameters of the upper and lower U-shaped rings at the bend after wearing for different wear times are described in Table 3. Figure 2 depicts that the wear rate of the upper and lower U-rings varies with the wear number. The wear rate of the upper U-shaped ring is higher than that of the lower one, which demonstrates that the wear of the upper U-shaped ring is more serious than that of the lower U-ring. This phenomenon is consistent with the actual transmission line wear failure. This is mainly because the microstructure in a state of plastic deformation alternately appears on both sides with the swinging of the lower U-shaped ring. Meanwhile, the contact area is in a state of plastic deformation for a long time, owing to the fixation of the upper U-shaped ring, which makes the surface of the upper U-shaped ring more likely to crack and fall off than the lower U-shaped ring.

It is seen in Figure 2 that there are two ways of increasing the wear rate, and the wear rate change can be divided into three stages under each wear load. The two growth methods are as follows: Under a small wear load, the difference in wear rate between the upper and lower U-shaped rings is not great with the increase of wear times. Under large wear loads, the difference between the upper and lower U-shaped ring wear rates increases with the increase of wear time. The three stages of wear rate change are II _Wear rate_ > III_Wear rate_ > I_Wear rate_. The reason for the different growth methods is that the contact area of the upper and lower U-shaped rings is mainly in elastic deformation under low wear loads. The contact area gradually changes from elastic deformation to elastic–plastic deformation along with the increase of wear load. During the subsequent periodic swinging of the U-shaped ring, the area in elastic–plastic deformation is prone to cracking and plastic collapse, which results in the increasing difference in wear rate.

### 3.2. Macroscopic Surface Morphology

The macroscopic morphology of the original sample, and the sample after 120,000 wear times, under different wear loads is shown in Figure 3. The original shape of the U-shaped ring is smooth (Figure 3a). The severe plastic flow zone was formed due to the long-term swinging of U-shaped ring, and the surface roughness increased at the wear load of 4000 N. There are spalling pits and very few shallow ploughs in the worn edge. This shows that adhesive wear mainly occurred at 4000 N. As the wear load increased, the surface roughness and the area of the plastic flow increased. Grooves, deep spalling pits, and cracks, formed along the swing direction of the U-shaped rings, could be observed at the wear edge, which indicates that as the wear load increased, not only the surface damage increased, but the wear mechanism also changed from adhesive wear, to severe adhesive and abrasive wear.

Elastic deformation mainly occurred in the contact area, in virtue of the small wear load. In the process of reciprocating swinging, local plastic deformation occurs on a small amount of asperity. The material is sheared into wear debris under the action of friction shear stress. Therefore, spalling pits left by adhesive wear could be observed on the surface at 4000 N. With the increase of contact pressure and friction heat between the upper U-shaped ring and lower U-shaped ring, the adhesion effect on the contact area was enhanced. A large amount of delamination produced pits, which aggravated the surface damage. Simultaneously, the exfoliating material was used as abrasive particles, and grooves occurred along the swing direction of the U-shaped rings [16].

### 3.3. Surface Damage

SEM micrographs and EDS composition analysis of the worn surfaces of the U-shaped ring under different wear loads are shown in Figure 4 and Table 4. It is noted that the surface damage morphology of the U-shaped rings varies with increasing wear loads. Figure 4a shows that the surface damage of the U-shaped ring presents spalling pits and slight ploughs. When the wear load increased to 6000 N, tiny delaminations and wear debris was observed, and plough effects were more serious. When the wear load further increased to 8000 N, the surface damage became more serious. The range and degree of delamination increased. Furthermore, cracks occurred in the delamination area. Meanwhile, the oxygen content increased as the wear load increased. The increase in carbon content was due to contamination of the conductive adhesive during EDS measurement.

Comprehensively analyzing macroscopic surface morphology and surface damage (Figure 3 and Figure 4), the damage mechanism transformed from adhesive wear, slight abrasive wear, and slight oxidation wear, to serious adhesive wear, abrasive wear, and oxidation wear, with increasing wear loads.

### 3.4. Metallography

The metallographic structure of the original sample is described in Figure 5. The U-shaped ring was composed of ferrite and pearlite, and the ferrite and pearlite were uniformly distributed without an obvious tendency. Under different wear loads, the microstructure of the U-shaped ring did not change, but the depth of the plastic deformation layer increased with the increase of wear load. In the plastic deformation layer, the microstructure showed this tendency. The softer ferrite was squeezed into a ferrite line along the sliding direction. When the residual stress in the plastic deformation zone was greater than the stress limit of the material, cracks were initiated in the softer ferrite. The cracks propagated along the ferrite line and terminated at the worn surface, which caused the worn surface material to exfoliate to form wear debris. With the increase of wear load and friction heat, the plastic deformation area and material damage of the U-shaped rings was aggravated. The macroscopic morphology was manifested as single shallow spalling pits and scratches, which gradually turned into pits and grooves [17].

### 3.5. TEM

TEM was used to further study the plastic deformation layer, as shown in Figure 6. The dislocations inside ferrite mainly existed in the form of a dislocation line in the original sample, and the dislocation density was low. In the 120,000 wear times, the dislocation density increased greatly and the dislocation became entangled inside the ferrite. Due to dislocation movement, dislocation pile-up appeared on the ferrite phase boundary. When the dislocation moved to the cementite, the dislocation was hindered by the cementite. With further aggravation of the plastic deformation, the cementite had local fractures.

### 3.6. Hardness

As shown in Figure 7, point A is the central region of the wear edge. Hardness at the cross section was measured throughout A–B–C, and the measurement results are shown in Figure 8. The higher the wear load, the lower the hardness value on the wear edge. As the hardness value of the first five points from the worn edge had not increased significantly, the average hardness values of the first five points were selected as the hardness of the worn edge. The edge hardness value of the wear load 4000 N was 198.35 HV, the edge hardness value of the wear load 6000 N was 171.91 HV, and the edge hardness value of the wear load 8000 N was 155.69 HV. At the wear edge, the dislocation disappeared due to an "outcrop", making the surface dislocation density lower than the subsurface dislocation density. Dislocation plugs easily form a cavity, and join together to form cracks in the area where the dislocation plug is deposited. The crack propagation along the ferrite line leads to the lower hardness of the wear edge under surface stress [18].

### 3.7. Wear Mechanism

The wear structure transformation of the U-shaped rings is shown in Figure 9. At the beginning of the wear, the upper U-shaped ring and lower U-shaped ring are in point contact, and there are a few dislocation lines inside the ferrite. With the application of wear load, the contact area between the two U-shaped rings increases, plastic deformation occurs in the surface structure of the contact area, dislocations are entangled with each other, and the ferrite turns fibrous. The structure shows a tendency along the sliding direction. As the wear progresses, the plastic flow increases, the cementite fractures, and cracks occur in the dislocation plugging area. The cracks propagate along the ferrite line and terminate at the worn surface, which causes the worn surface material to exfoliate to form a spalling pit. The larger the wear load, the more plastic flow and damage are aggravated on the surface of the material. The wear debris is used as abrasive particles to scratch the surface of the material and cause ploughing, which leads to the transformation of the U-shaped ring wear mechanism under different loads [19].

## 4. Wear Simulation Analysis

### 4.1. Theoretical Basis of Wear Simulation

The Archard wear model is the most widely used in wear simulation [20]
(1)$$\frac{dv}{ds}=K\frac{F}{H}$$
where dv is the wear volume, ds is the sliding distance, K is the wear coefficient, F is the applied load, and H is the material hardness (N/m^2^).

The left and right sides of the equal sign in Equation (1) are divided by a tiny contact area, ΔA, to get the generalized Archard wear model [21].
(2)$$\frac{dh}{ds}=KdP$$

Where dh is the wear depth,Kd is the line wear coefficient, and P is contact pressure.

Assuming that the contact node of the mesh between the upper and lower U-rings is i after the mesh is divided, and the entire wear process is discretized into j wear steps. Then from Equation (2), the wear depth of the i contact node at the j wear step is as follows [22].
(3)$$\Delta h_{i,j}=\Delta s_{i,j}\cdot K_{di,j}\cdot P_{i,j}$$

Based on the discrete equation of wear, a Umeshmotion wear subroutine for connection fittings was written in Fortran language [23,24]. The Arbitrary Lagrangian Eulerian adaptive mesh update technology in Abaqus was used to realize the automatic update of the mesh nodes of the worn surface after each wear step, so as to simulate the wear of the U-shaped ring during the mutual sliding process [25]. The wear simulation process is shown in Figure 10.

### 4.2. Wear Model

According to the above wear test, the wear degree of the upper U-shaped ring was greater than that of the lower U-shaped ring; therefore, only the wear depth of the upper U-ring is simulated below. In order to facilitate the application of boundary conditions, on the premise of not affecting the simulation results, the U-shaped ring was appropriately simplified. The dimensions of the model were consistent with the U-shaped ring in the tests. The model was drawn in UG NX 10.0 (Siemens, Munich, Germany), and shown in Figure 11. The performance parameters of the U-shaped ring are shown in Table 5. Two analysis steps were set. The mesh type was C3D8, with 9945 meshes divided by the upper U-shaped ring, and 1728 meshes by the lower U-shaped ring. Full restraint was imposed on both ends of the upper U-shaped ring. The wear load was 6000 N, the lower U-shaped ring swung around its own axis by 30°, and the amplitude periodic function was θ = 0.1666667sin(2πt). The boundary conditions were set as shown in Figure 12. $K_{d}$ = 9.779 × 10^−8^ mm^3^/N·mm.

## 5. Simulation Results and Analysis

### 5.1. Wear Depth Simulation

A workstation was used to simulate the wear depth of the upper U-shaped ring, and the comparison between the simulation results and the test values is shown in Table 6 and Figure 13. The simulated value was consistent with the growth trend of the experimental value, and the average error between the simulated value and the experimental value was 1.56%, which is in good agreement. This proves that the numerical simulation method can be effectively used in the wear research of the connection fitting. Figure 14 shows the simulation results of wear depth for 20,000 times, 30,000 times, 50,000 times, and 70,000 times, and the comparison between the simulated total deformation of 80,000 times, and the sample after wearing 80,000 times. It can be seen from Figure 13 that the wear depth and wear area of the U-shaped rings gradually increased with the increase of wear time, and the maximum wear depth occurred in the central area of contact. This is consistent with the physical failure and wear sample. Moreover, as the wear number increased, the upper U-shaped ring was gradually stretched due to the ratchet effect.

### 5.2. Wear Affected Area

The number of meshes updated on the worn surface was used to characterize the affected area, as shown in Figure 15. As the amount of wear increases, the area affected by wear increases rapidly at the beginning. The growth of the affected area gradually slowed down after 40,000 wear times. After 80,000 wear times, the growth rate of the affected area further slowed down. The wear depth was the highest in the central area of wear. As the distance from the wear central area increased, the wear depth of the node gradually decreased (Figure 14). In order to elucidate the change rule of wear depth on the worn surface, the C and D paths are shown in Figure 16, to read the wear depth of the nodes on these two paths. The change of wear depth of the nodes is shown in Figure 17. To begin with, the wear widths on path C were much larger than those on path D. With the increase of wear times, the wear widths in path C and D gradually approached each other. According to Figure 13 and Figure 17, the following three stages of wear rate change were refined. In the first stage, between 0 to 40,000 wear times, the area affected by wear increases rapidly, and the wear depth gradually increases. In the second stage, the growth of the wear affected area slows down, and the wear depth increases rapidly between 40,000–80,000 wear times. In the third stage, the growth of the wear affected area is further slowed down, between 80,000 to 120,000 times. The decline of wear depth is mainly concentrated on both sides of the wear central area; hence the growth of wear depth is slowed down.

## 6. Conclusions

The following conclusions are based on the combined computational and experimental studies on U-shaped ring wear under different wear loads.

In strong wind environments, the wear depth of upper and lower U-shaped rings increases gradually with the increase of wear times, and the wear depth of the upper U-shaped ring is more serious than that of the lower U-shaped ring. There are two ways of increasing wear rate, the wear rate fluctuations can be divided into three stages under different wear loads, II _Wear rate_ > III_Wear rate_ > I_Wear rate_.The wear mechanism transforms from adhesive wear, slight abrasive wear, and slight oxidation wear, to serious adhesive wear, abrasive wear, and oxidation wear with the increase of wear load. Meanwhile, with the increase of wear load, the surface damage is aggravated, the hardness of the wear edge decreases, and the plastic deformation area increases. In the plastic deformation area, dislocation plugging occurs at the ferrite phase interface, and cementite fracture occurs.The error between the test value and the simulation value of the wear depth was 1.56%, which proves the reliability of the established wear failure prediction model for power connection fittings.

## Figures and Tables

**Figure 1 materials-14-00735-f001:**
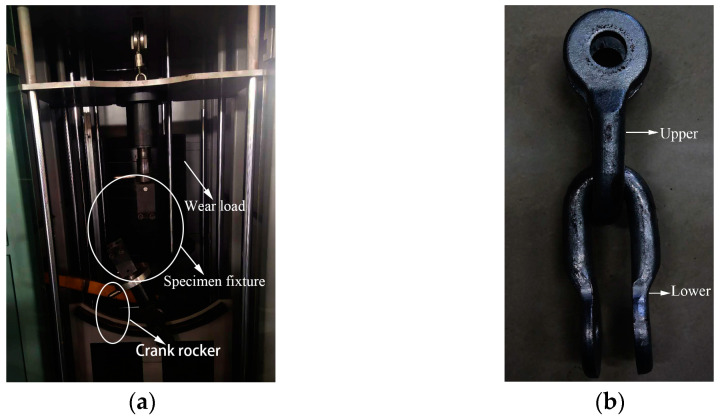
Upper and lower U-shaped ring, (**a**) wear tester; (**b**) U-shaped ring assembly image.

**Figure 2 materials-14-00735-f002:**
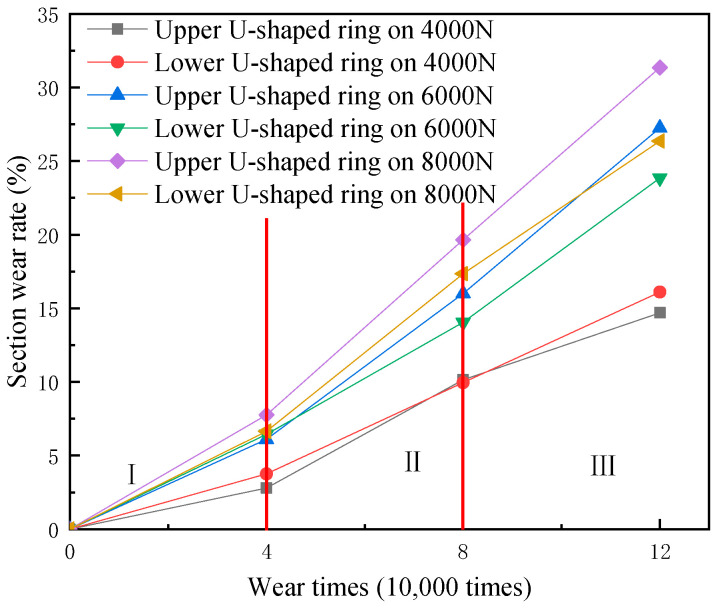
Change graph of wear rate of the upper and lower U-shaped ring sections with wear times.

**Figure 3 materials-14-00735-f003:**
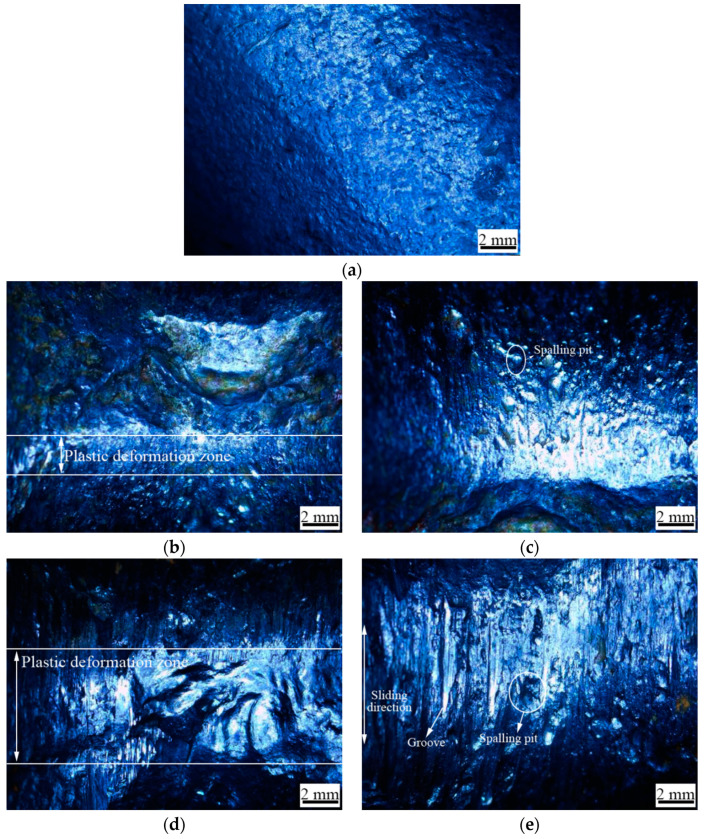
Surface morphology of the worn surface, (**a**) the original sample; (**b**)wear load 4000 N; (**c**) wear edge on wear load 4000 N; (**d**) wear load 6000 N; (**e**) wear edge on wear load 6000 N; (**f**) wear load 8000 N; (**g**) wear edge on wear load 8000 N.

**Figure 4 materials-14-00735-f004:**
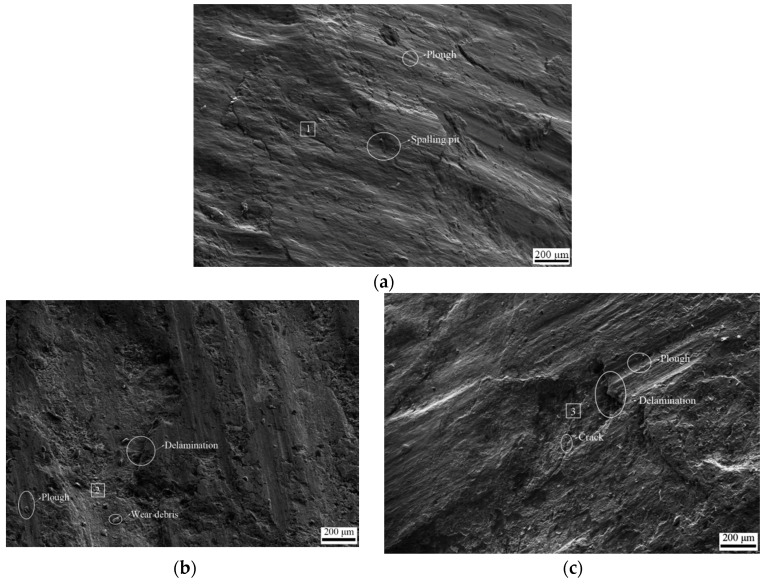
SEM micrographs of surface damage of the U-shaped rings, (**a**) wear load 4000 N; (**b**) wear load 6000 N; (**c**) wear load 8000 N. 1: area 1; 2: area 2; 3: area3.

**Figure 5 materials-14-00735-f005:**
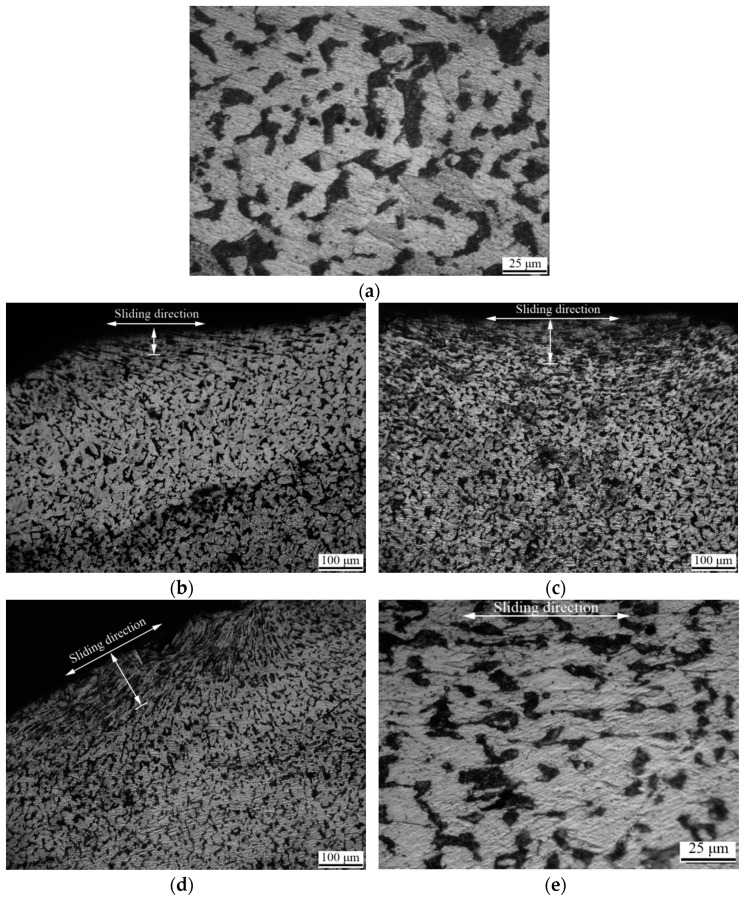
Wear edge metallographic structure under different wear loads, (**a**) the original sample; (**b**) wear load 4000 N; (**c**) wear load 6000 N; (**d**) wear load 8000 N; (**e**) enlarged view of wear load 8000 N.

**Figure 6 materials-14-00735-f006:**
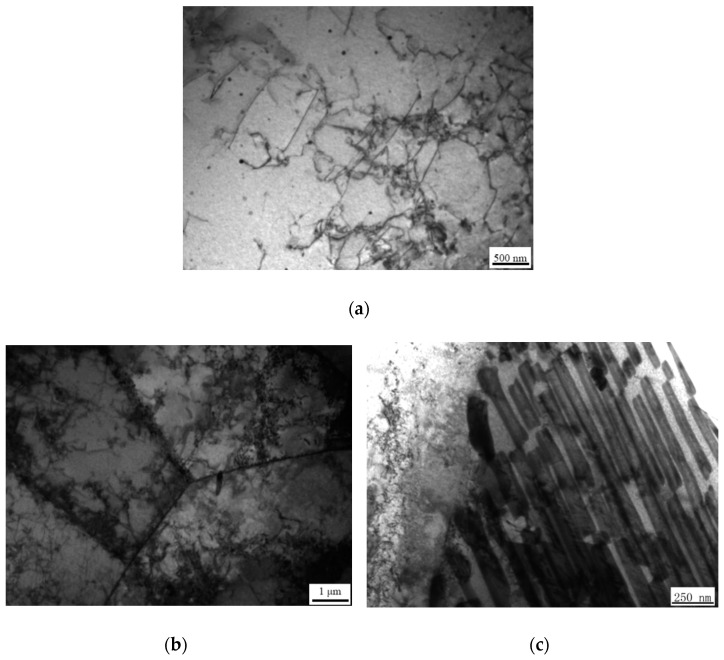
TEM image of the worn edge, (**a**) the original sample; (**b**) ferrite, (**c**) pearlite.

**Figure 7 materials-14-00735-f007:**
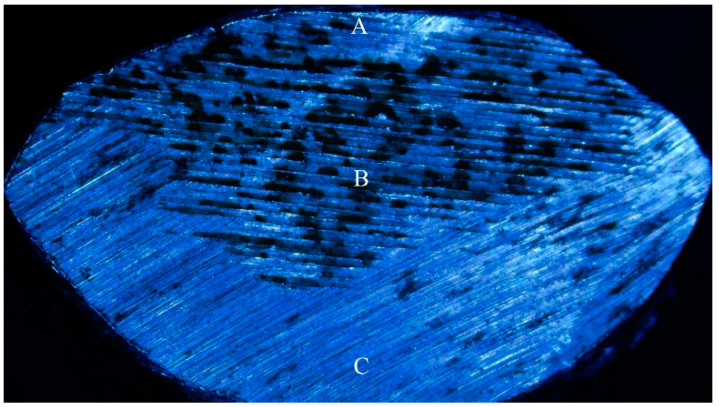
A-B-C.

**Figure 8 materials-14-00735-f008:**
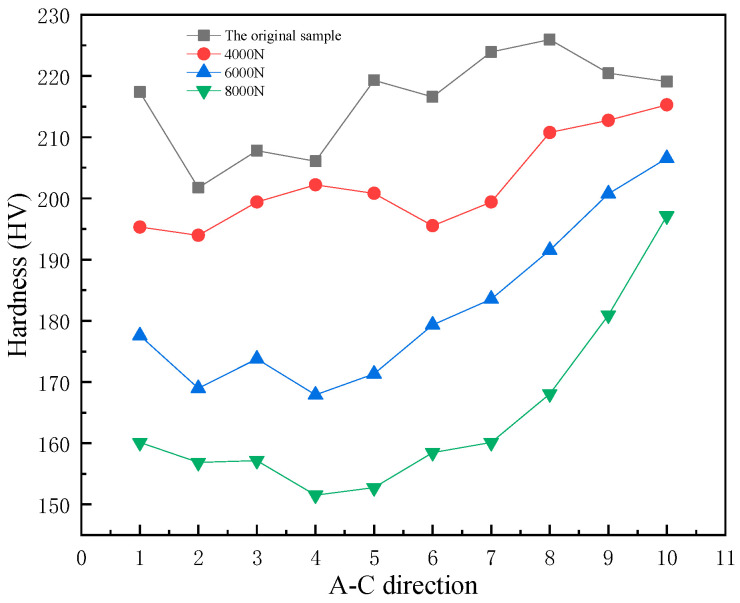
A-C hardness change.

**Figure 9 materials-14-00735-f009:**
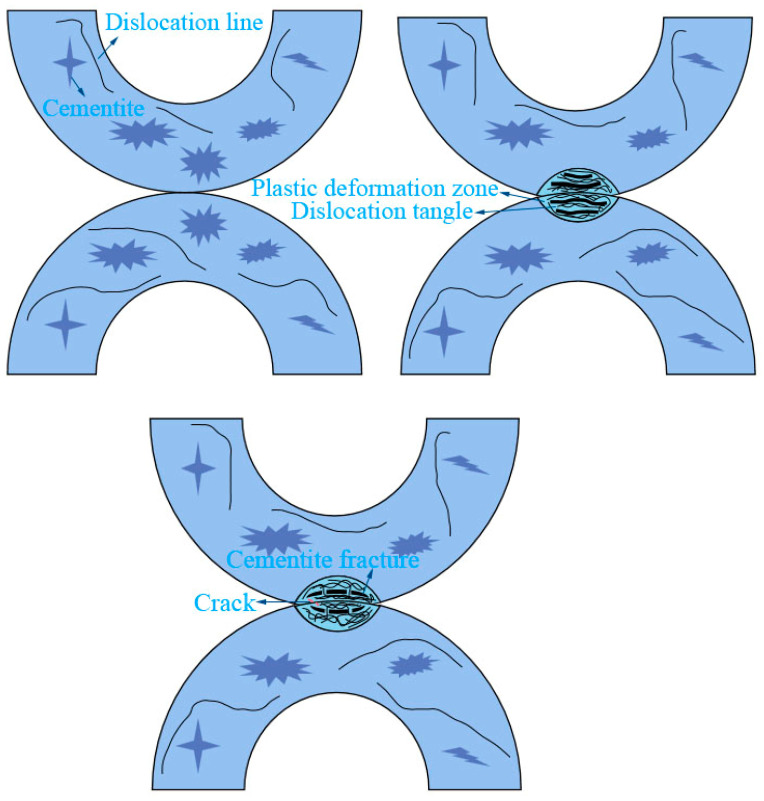
Wear structure transformation of the U-shaped ring.

**Figure 10 materials-14-00735-f010:**
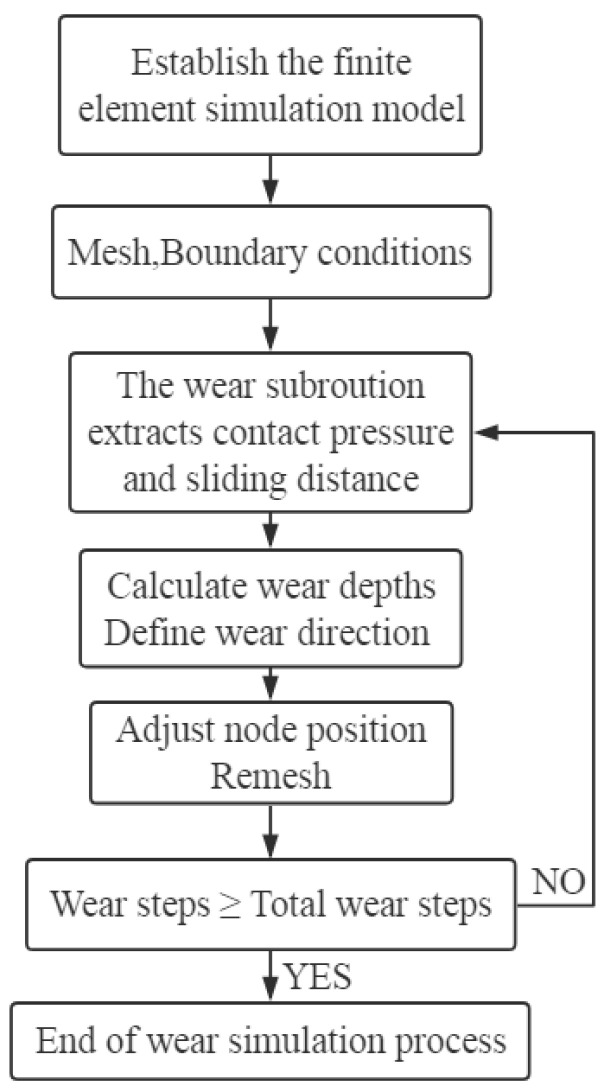
Flow chart of wear simulation.

**Figure 11 materials-14-00735-f011:**
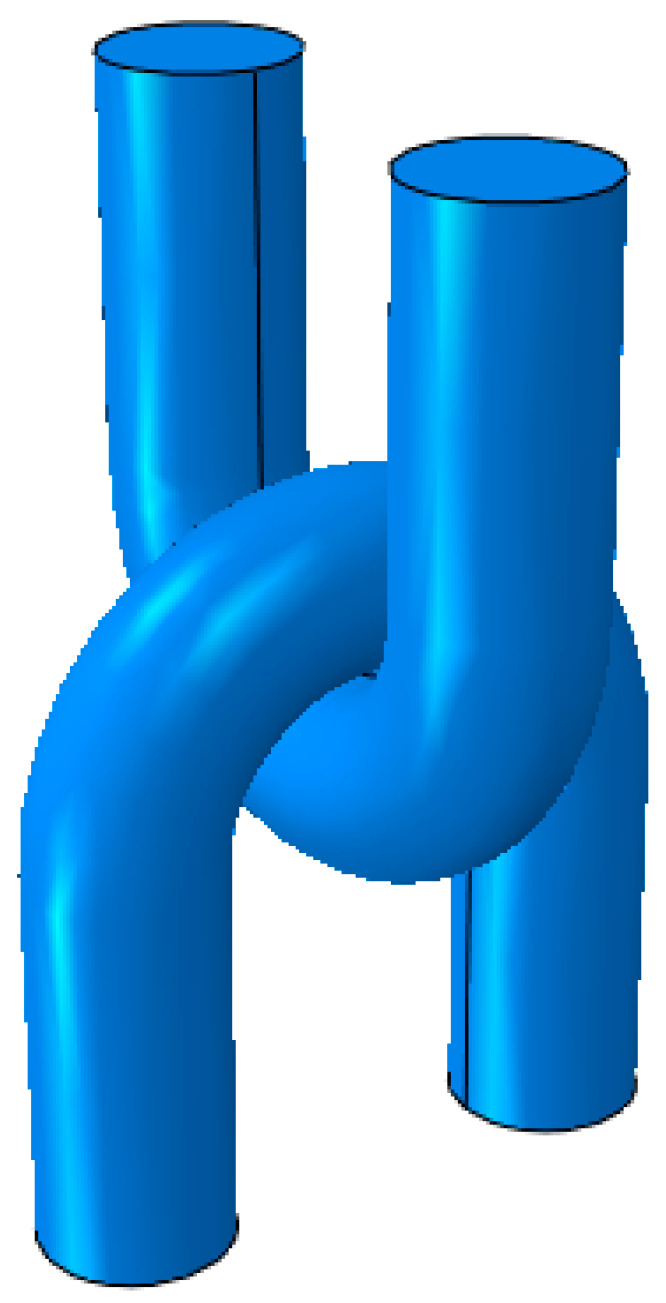
U-shaped ring finite element model.

**Figure 12 materials-14-00735-f012:**
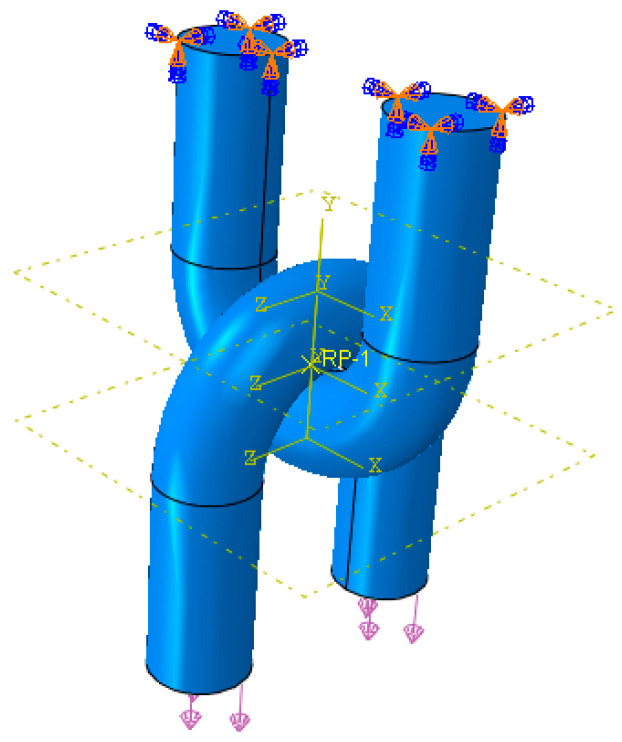
U-shaped ring imposed boundary conditions.

**Figure 13 materials-14-00735-f013:**
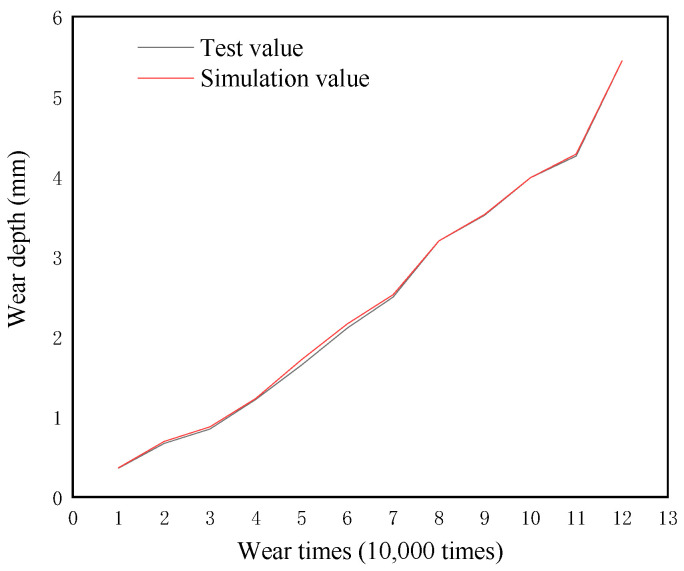
Comparison of wear depth between test values and simulation values.

**Figure 14 materials-14-00735-f014:**
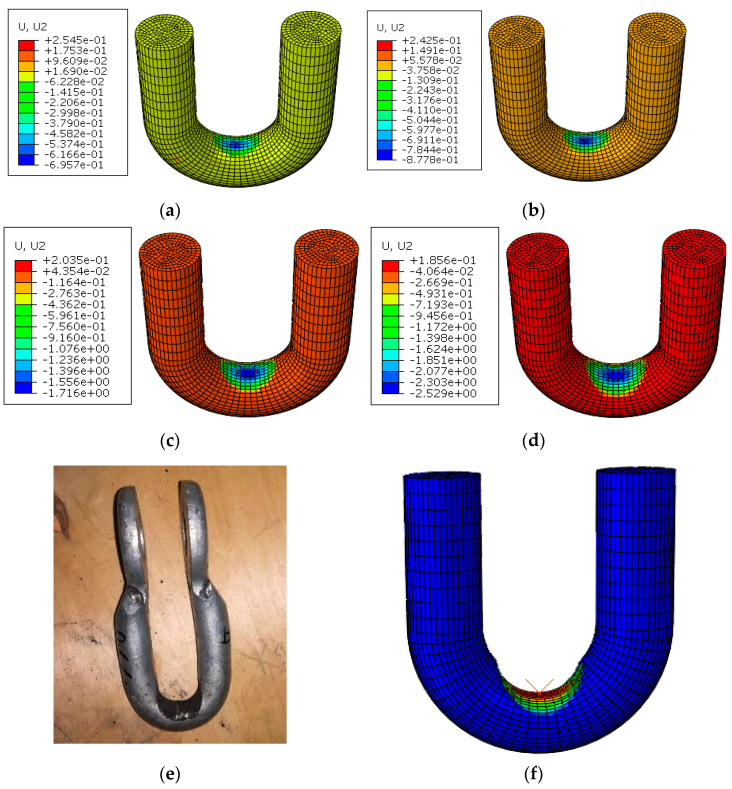
Change of wear depth with wear time, (**a**) 20,000 times; (**b**) 30,000 times; (**c**) 50,000 times; (**d**) 70,000 times; (**e**) 80,000 times; (**f**) total deformation at 80,000 times.

**Figure 15 materials-14-00735-f015:**
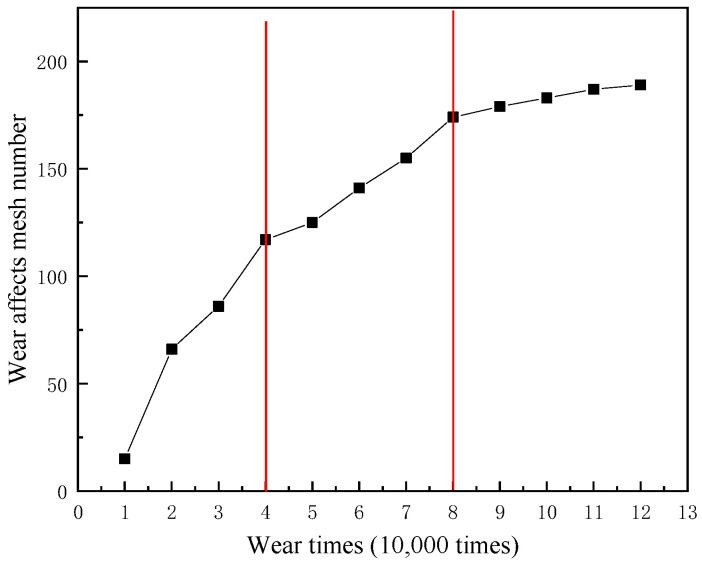
Change of wear area with wear times.

**Figure 16 materials-14-00735-f016:**
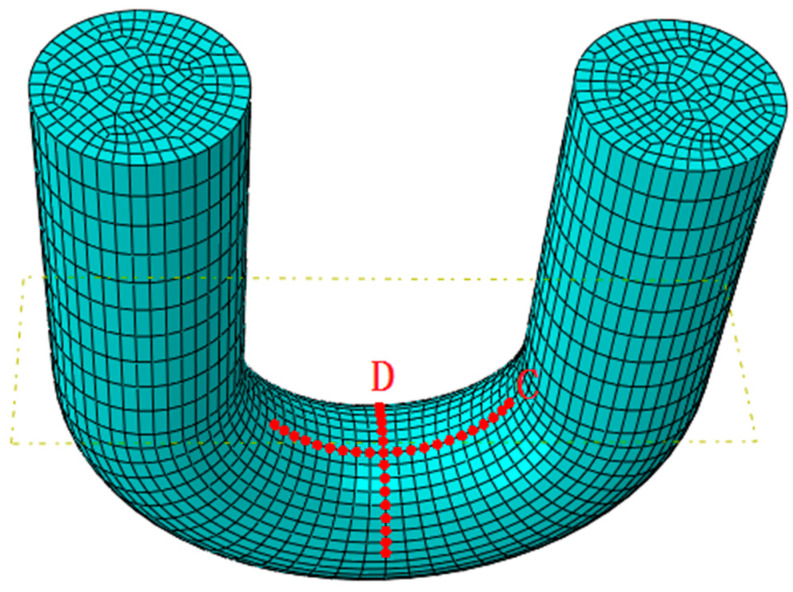
Path.

**Figure 17 materials-14-00735-f017:**
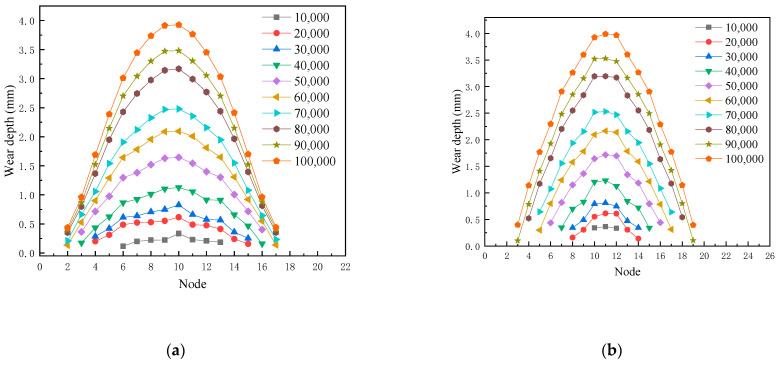
Variation of node wear depth for different paths, (**a**) C path; (**b**) D path.

**Table 1 materials-14-00735-t001:** Chemical composition of the U-shaped ring (wt.%).

Chemical Composition				
Fe	C	Mn	Si	S
99.30	0.16	0.30	0.20	0.04

**Table 2 materials-14-00735-t002:** Designed parameters of the Tu-ha Line II.

Project	Ground Wire	Optical Cable
Total horizontal load of wire/N	6931	7731
Total vertical load of wire/N	4786	5275
Comprehensive wire load/N	8423	9359
Safety factor of hanging wire fittings	14	13
Wind deflection (gale) degrees	55.33	55.68

**Table 3 materials-14-00735-t003:** Wear test results of upper and lower U-shaped rings.

**Wear** **Load**	**Wear Times**							
	0 time		40,000 times		80,000 times		120,000 times	
	upper/mm	lower/mm	upper/mm	lower/mm	upper/mm	lower/mm	upper/mm	lower/mm
4000 N	20	20	19.44	19.25	17.97	18.01	17.06	16.78
6000 N	20	20	18.78	18.71	16.8	17.19	14.55	15.23
8000 N	20	20	18.45	18.67	16.07	16.53	13.73	14.73

**Table 4 materials-14-00735-t004:** EDS composition analysis of the worn surface under different wear loads (wt.%).

Area	Fe	O	C	Mn	Si	S
1	79.96	12.09	7.53	0.17	0.18	0.06
2	69.72	21.66	7.86	0.09	0.65	0.02
3	62.01	33.16	4.37	0.3	0.13	0.03

**Table 5 materials-14-00735-t005:** Performance parameters of the U-shaped ring.

Density/(g/cm^3^)	Elastic Modulus/MPa	Poisson’s ratio	Yield Strength/MPa
7.85	210,000	0.3	235

**Table 6 materials-14-00735-t006:** Comparison of simulated and experimental values with the number of wear at 6000 N.

Wear Times (Ten Thousand Times) Wear Loads are 6000 N	The Test Results	Simulation Value
0	0	0
1	0.036	0.3675
2	0.57	0.6957
3	0.85	0.8778
4	1.22	1.233
5	1.65	1.716
6	2.11	2.164
7	2.50	2.529
8	3.20	3.300
9	3.52	3.529
10	3.99	3.990
11	4.26	4.286
12	5.45	5.452

## Data Availability

Data sharing not applicable.

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
