# Peer review of "Simulation and Experimental Study on Wear of U-Shaped Rings of Power Connection Fittings under Strong Wind Environment"

_materials, 2021, doi:10.3390/ma14040735_

Round 1
Reviewer 1 Report
- What newest data is in this manuscript?
- The keywords includ "surface topography", which had not been studied.
- There is no detailed description of research devices as well as tribological tests.
- Table 2. What do the JLB20A-100 and OPGW0115 mean?
- Table 3. Data in the table (first row) are incomprehensible - what is what?
- How many tribological tests have been performed? The graphs show that only one - see fig.13 and fig.15.
- Paragraph 3.2, lines 127-130. The authors wrote: "... is relatively smooth and surface roughness is small ...". On what basis such a coment? In order to assess the surface topography and comment to the the surface roughness, measurements should be made on appropriate devices (for example profilometer, interferometer or confocal microscope) and roughness parameters should be generated (example, Sq, Sz, Ssk, Sku, etc.).
This is not in the manuscript. - The quality of the markings in the images is poor - see figures fig.1, fig.3, fig.4, fig.5, fig.7.
- Figure 17. There is no information, that the legend presents different wear time.
Author Response
Dear reviewer:
Thank you for the comments concerning our manuscript entitled “Simulation and Experimental Study on Wear of U-shaped Rings of power Connection Fittings under Strong Wind Environment” (ID:materials-1070637). Those comments are all valuable and very helpful for revising and improving our paper, as well as the important significance to our researches. We have studies comments carefully and have made correction which we hope meet with approval. Please see the attachment.

Reviewer 2 Report
The topic of the manuscript entitled ‘Simulation and experimental study on wear of U-shaped rings of power connection fittings under strong wind environment’ falls within the scope of Materials. In the manuscript, the wear evolution and failure mechanism of U-shaped ring with different wear loads were tested and based on Archard wear model was simulated in ABAQUS program. The paper contains interesting experimental results and corresponding analyses. It is of sufficient scientific interest and has originality in its technical content to merit publication. The authors have cited the relevant literature. Methods and interpretations of results are correct and novel. The issues were well presented. The arrangement of work maintains substantive continuity and constitutes a logical whole. References are not prepared in accordance with the Instructions for authors and template.
The manuscript (after improvement References) is suitable for publication in its present form.
Author Response

(The authors gave the same response as above.)

Reviewer 3 Report
The manuscript titled “Simulation and Experimental Study on Wear of U-shaped Rings 2 of power Connection Fittings under Strong Wind Environment” is presenting investigations on the effect of applied load during wear tests on unspecified material U-shaped rings.
This paper is not recommended for publication in the Materials journal, in the present form.
Following are only some of the issues in the manuscript which are to be seriously addressed for accepting its publication in the journal:
- The English language and grammar used in the present manuscript is generally poor. There are plenty of instances where mistakes, misspelled words and/or poorly chosen words (ambiguous), and unnecessary repetitions are present. I strongly suggest that the paper should be proofread and double-checked concerning the spelling and phrasing. This version is very difficult to read and to understand.
- Considering that the journal topic concerns the “materials” aspect, minimal importance was given by the authors to this aspect. I am guessing that the authors are speaking about a cast iron, or maybe a steel grade (although the carbon content is huge). I don’t understand what is the relation between this manuscript with the journal subject matter. The manuscript is a very particular case study, and less so a scientific article. The novelty, especially regarding the “materials” aspect, is not mentioned by the authors. I had to look on the internet the meaning of Q235, which seems to be a low carbon steel. This information should have been given in the manuscript.
- Table 2 is not clear, especially the last two columns (unexplained information, what is JLB20A-100 or OPGW-115). Error values are missing throughout the manuscript. The images from figures 3 and 4 are barely legible. Consequently, one has to trust that the discussions regarding the wear mechanisms are adequate. Table 4, the variation in composition is not explained (huge quantity of oxygen and carbon), especially if the manuscript is about a low carbon steel. Most of the discussions are superficial.
Author Response

(The authors gave the same response as above.)

Reviewer 4 Report
Please make a space between text and citations (ex. 1Hz [15]) in all the document
In fig. 1. a, please change the color of the text (the red color is not so clear)
Figure 1. Upper and lower U-shaped ring, (a) wear tester; (b) U-shaped ring assembly drawing (some spaces are needed)
In table 1 the units of the density is better to be, g/cm3
At 4.1 Theoretical basis of wear simulation, line 250 - ? where is the wear...., the same at line 254
The model used for finite element method presents the same dimensions as the U-shaped used at tests? The model was made in Abaqus or other software?
Author Response

(The authors gave the same response as above.)

Round 2
Reviewer 1 Report
The manuscript seems to be much better than its previous version.
I have one main comment - please change the "surface topography" into "surface morphology" in keywords and the title of Figure 3 (it should be "Surface morphology" not "Macroscopic topography" !).
Afetr these correction, the manuscript can be published in the Journal.
Author Response
Dear reviewer:
Thank you for the comments concerning our manuscript entitled “Simulation and Experimental Study on Wear of U-shaped Rings of power Connection Fittings under Strong Wind Environment” (ID:materials-1070637). Those comments are all valuable and very helpful for revising and improving our paper, as well as the important significance to our researches. We have studies comments carefully and have made correction which we hope meet with approval.The main corrections are as following:
Comment 1: The manuscript seems to be much better than its previous version.
I have one main comment - please change the "surface topography" into "surface morphology" in keywords and the title of Figure 3 (it should be "Surface morphology" not "Macroscopic topography" !).
Response 1: Thank you very much for your valuable comment. We have changed the expression in lines 21, 156.